# Risk of Hepatitis E among Persons Who Inject Drugs in Hong Kong: A Qualitative and Quantitative Serological Analysis

**DOI:** 10.3390/microorganisms8050675

**Published:** 2020-05-06

**Authors:** Siddharth Sridhar, Nicholas Foo-Siong Chew, Jianwen Situ, Shusheng Wu, Ernest Sing-Hong Chui, Athene Hoi-Ying Lam, Jian-Piao Cai, Vincent Chi-Chung Cheng, Kwok-Yung Yuen

**Affiliations:** 1Department of Microbiology, Li Ka Shing Faculty of Medicine, The University of Hong Kong, Hong Kong, China; chewnf@hku.hk (N.F.-S.C.); situjw@hku.hk (J.S.); wss2017@hku.hk (S.W.); ernesto@connect.hku.hk (E.S.-H.C.); athenelx@connect.hku.hk (A.H.-Y.L.); caijuice@163.com (J.-P.C.); vcccheng@hku.hk (V.C.-C.C.); 2State Key Laboratory of Emerging Infectious Diseases, The University of Hong Kong, Hong Kong, China; 3Carol Yu Centre for Infection, The University of Hong Kong, Hong Kong, China; 4The Collaborative Innovation Center for Diagnosis and Treatment of Infectious Diseases, The University of Hong Kong, Hong Kong, China

**Keywords:** hepatitis E, seroepidemiology, persons who inject drugs

## Abstract

Hepatitis E virus (HEV) is an important cause of hepatitis, which can be transmitted via the bloodborne route. However, risk of hepatitis E among persons who inject drugs (PWIDs) is poorly understood. This study aimed to elucidate whether PWIDs are at risk for hepatitis E. We performed HEV IgM, IgG and nucleic acid detection on a cohort of 91 PWIDs and 91 age- and sex-matched organ donors. Blood HEV IgG was measured using the WHO HEV antibody standard. The effects of age, gender and addictive injection use on HEV serostatus and concentration were assessed. HEV IgG seroprevalence was 42/91 (46.2%) in the PWID group and 20/91 (22%) in the donor group (odds ratio = 3.04 (1.59–5.79), *p* = 0.0006). The median HEV IgG concentration was 5.8 U/mL (IQR: 2.5–7.9) in the PWID group and 2.1 U/mL (IQR: 1.2–5.3) in the donor group (*p* = 0.005). Increasing age and addictive injection use were significantly associated with HEV IgG serostatus, but only addictive injection use was associated with HEV IgG concentration (*p* = 0.024). We conclude that PWIDs are at increased risk for hepatitis E and are prone to repeated HEV exposure and reinfection as indicated by higher HEV IgG concentrations.

## 1. Introduction

Hepatitis E virus (HEV) is an important cause of viral hepatitis globally [1]. In humans, hepatitis E is mostly caused by four genotypes within species *A* of the *Orthohepevirus* genus (HEV-A) under the family *Hepeviridae* [2]. HEV-A genotypes 1 and 2 are spread between humans via the feco-oral route and circulate in areas with lower socioeconomic development [1]. HEV-A genotype 3 (Europe and the Americas) and genotype 4 (China) are mostly transmitted from pigs to humans as a foodborne zoonosis [3,4,5]. In addition, we recently discovered that *Orthohepevirus* species *C* genotype 1 (HEV-C1), also known as rat hepatitis E, can cause hepatitis in humans [6,7].

In addition to foodborne transmission, HEV-A genotypes 3 and 4 can also be transmitted via contaminated blood products or organs [8,9]. Asymptomatic viremic blood donors have been documented in several countries [10]. Such bloodborne transmission has prompted several countries to initiate HEV screening of blood and organ donors [11,12].

Bloodborne infections are common among people who inject drugs (PWIDs) due to high-risk behavior such as needle sharing. However, the importance of addictive injection behavior as a risk factor for hepatitis E is uncertain. HEV IgG seroprevalence among PWIDs in Europe and North America ranges from 2.8–62% [13,14,15,16,17,18,19,20]. Comparisons of the HEV seroprevalence between PWIDs and non-drug-using control groups have produced conflicting results; some studies have found significant differences [18,19,21], while others have concluded that PWIDs are not at increased risk of hepatitis E [13,15,16]. Most studies had significant methodological issues such as small sample sizes [18,22,23], unmatched control populations [17,18,20,21,23,24], or control populations that are not representative of the general population such as hepatitis C virus (HCV) carriers, prisoners or homeless persons [14,18,20,23]. None of the studies quantified HEV IgG in sera of study subjects. 

In this study, we investigated the association between addictive injection use and hepatitis E by conducting a matched cohort study among PWIDs and organ donors in Hong Kong, a HEV-A genotype 4 endemic area with a population HEV seroprevalence of 15.8% [25,26]. Both qualitative seroprevalence and HEV IgG concentrations were compared between groups.

## 2. Materials and Methods

### 2.1. Patients and Controls

The study was conducted in the Department of Microbiology of Queen Mary Hospital, which provides diagnostic testing for organ transplant centers and viral hepatitis clinics throughout Hong Kong. Archived plasma samples from adult PWIDs with known chronic HCV infection who sent blood to the laboratory for HCV load testing between 1 January 2018 to 31 October 2019 were retrieved. We evaluated HCV-infected PWIDs because this is an indicator of high-risk practices such as needle sharing [27,28], which in turn would render these individuals at higher risk of other bloodborne infections like hepatitis E. Both current drug users and people who had previously used injection drugs were included. Subjects were excluded if they were co-infected with HIV, were men who have sex with men, had received blood transfusions or were under any form of immunosuppression.

PWIDs were individually age- and sex-matched in a 1:1 ratio with potential organ donors who sent sera to the laboratory for pre-donation bloodborne virus screening. All organ donors tested negative for HCV and HIV antibodies. None of them had a history of addictive injection use. All PWIDs and organ donors were permanent residents of Hong Kong. This study was approved by the Institutional Review Board of the University of Hong Kong/Hospital Authority Hong Kong West Cluster (UW 18-074) on 17 December 2019.

### 2.2. Hepatitis E Serology

Hepatitis E IgM and IgG testing were performed for all PWID and organ donor blood samples using Wantai immunoassay kits (Wantai, Beijing, China) as per the manufacturer’s instructions. HEV IgG in blood was quantified using the WHO reference reagent for hepatitis E virus antibody (NIBSC: code 95/584, Potters Bar, UK) as described previously by Abravanel et al. [29]. The linearity of the Wantai IgG assay was verified by testing five replicates of four dilutions of the NIBSC standard ranging from 0.625–5 U/mL. Subsequently, for assay runs involving PWID and organ donor blood samples, we tested two replicates of each of these five dilutions to obtain a standard curve. For blood samples testing positive for HEV IgG according to the assay criteria, optical density values were converted into U/mL using the standard curve. Blood samples with HEV IgG > 5 U/mL were re-tested after dilution to obtain the IgG concentration.

### 2.3. Hepatitis E Nucleic Acid Detection

Blood samples from PWIDs and organ donors were pooled together in groups of 10 or 11. Total nucleic acid was extracted from 200 µL of these minipools using the EZ1 Virus Mini Kit v2.0 (Qiagen, Hilden, Germany). In-house-developed HEV-A and HEV-C1 specific real-time reverse-transcription polymerase chain reaction (RT-PCR) assays were performed for each minipool, as previously described [6]. The analytical sensitivity of HEV-A and HEV-C1 RT-PCR assays were 33 (95% CI: 21–79) copies/reaction and 34 (95% CI: 21–116) copies/reaction, respectively.

### 2.4. Statistical Analysis

Non-normal and censored continuous variables were described in terms of median and interquartile range (IQR). The Student’s *t*-test was used to compare means of log-transformed HEV concentration data. The chi-square test was used to compare proportions of HEV seropositive individuals in various age brackets, gender groups and according to injection use status. Multiple regression methods (logistic and ANCOVA) were used to model the effect of age as a quantitative variable and injection use status on HEV IgG seroprevalence and concentration. Statistical analysis was performed using XLSTAT (Addinsoft, Long Island, NY, USA) and GraphPad Prism Version 8.1 (GraphPad Software, La Jolla, CA, USA). Post-hoc power analysis was performed using an online clinical calculator (https://clincalc.com/stats/power.aspx).

## 3. Results

### 3.1. Characteristics of PWIDs and Organ Donors

We included 91 HCV-infected PWIDs in this study and matched them according to age and sex to 91 organ donors. Seventy-five out of 91 (82.4%) of each group were male. Their median age was 46 (IQR: 41–55). None of the study subjects had a documented history of hepatitis E in the past. In the PWID group, 82/91 (90.1%) were ethnic Chinese compared to 89/91 (97.8%) in the donor group (*p* = 0.058). None of the patients in the PWID group were taking direct-acting antivirals, although 3/91 (3.3%) had received or were receiving ribavirin and pegylated interferon for treatment of hepatitis C. The HEV IgG seroprevalence in the PWID group was 42/91 (46.2%) and in the donor group was 20/91 (22%); this difference was statistically significant using the Chi-square test (*p* = 0.0006). One PWID but none of the donors tested positive for HEV IgM. The HEV IgM positive PWID had mild biochemical hepatitis on day of blood taking (alanine aminotransferase: 80 IU/L), but was otherwise asymptomatic. His HEV IgG was positive (1.2 U/mL). None of the organ donor or PWID blood minipools tested positive for HEV-A or HEV-C1 RNA. Cycle threshold values of plasmid standards and positive controls were satisfactory indicating validity of RT-PCR runs.

### 3.2. HEV IgG Quantitation

The Wantai IgG assay was confirmed to be linear in the range 0.625–5 U/mL (Figure 1). HEV IgG was quantified in the blood samples of 42 PWIDs and 20 organ donors testing positive by the Wantai HEV IgG assay (Figure 2). The median concentration of HEV IgG antibodies in the PWID group was 5.8 U/mL (IQR: 2.5–7.9) and in the donor group was 2.1 U/mL (IQR: 1.2–5.3). Comparison using the Mann–Whitney U test showed significant difference between the median of the two groups (*p* = 0.005). The antibody concentrations were log_10_-transformed for normality (verified by the Shapiro–Wilk test) and both groups were then compared using the Student’s *t*-test, which showed that the difference between log-transformed means was significantly different (*p* = 0.004).

### 3.3. Bivariable and Multiple Regression Analyses

The effects of age, gender and addictive injection behavior on hepatitis E seroprevalence and IgG concentrations were evaluated for the 182 PWIDs and organ donors included in this study. On bivariable analysis, age > 54 years and addictive injection use were significantly associated with HEV IgG serostatus (Table 1). Based on this, we constructed a multiple logistic regression model including age and addictive injection use as potential explanatory variables for the HEV IgG serostatus of 182 PWIDs and donors. As shown in Table 1, both variables were included in the final model (*p* < 0.005). For the 62 PWIDs and donors with detectable HEV IgG, we constructed a multiple regression model with HEV IgG concentration as the dependent variable; age and addictive injection use were used as explanatory variables. Interestingly, only addictive injection use (*p* = 0.024) was significant in the final model. Age was not significantly associated with HEV IgG concentration (*p* = 0.817).

## 4. Discussion

Previous studies examining the impact of addictive injection use on hepatitis E risk have produced contradictory results. Two studies found evidence for increased risk of hepatitis E among PWIDs [18,19]. However, the control populations in most studies were unmatched to the PWID groups, even for age, which is a major determinant of HEV seropositivity [30,31]. Only one study included an age- and sex-matched control population [13]. However, a limitation of this study was its small sample size of only 52 PWIDs. The findings of previous studies are summarized in Table 2.

Our study featured methodological improvements on these studies. We included a matched control population, had adequate sample size (post-hoc power calculation = 93.8%), quantified HEV IgG and selected PWIDs who had contracted HCV previously (indicating higher risk of needle-sharing and bloodborne viral infections). Furthermore, to the best of our knowledge, this is the first study of hepatitis E among PWIDs in an East Asian HEV-A genotype 4 endemic setting. HEV-A genotype 4 accounts for 76% of virologically confirmed hepatitis E cases in Hong Kong [7]. Like HEV-A genotype 3, genotype 4 is predominantly a foodborne zoonosis acquired by the consumption of undercooked pork products [26]. However, we have documented nosocomial transmission of genotype 4 in the transplantation setting [9]. Therefore, bloodborne transmission of this genotype in PWIDs is possible.

In this study, we found that PWIDs had significantly higher HEV IgG seroprevalence than organ donors (representing the general population). Our estimate of HEV seroprevalence in the general population is similar to that of a recent study conducted among blood donors in Hong Kong [25]. Furthermore, PWIDs had higher circulating HEV IgG concentration than organ donors. Pathogen-specific IgG is produced upon first exposure to viral infections; levels typically peak within a few weeks of infection and then gradually decline. Exposure to another strain of the same virus produces a robust spike in pathogen specific IgG. Therefore, the finding of elevated HEV IgG in PWIDs suggests that they were exposed to HEV repeatedly, resulting in robust anamnestic IgG responses. Evidence of repeated exposure is a particularly important finding given the mounting evidence that HEV reinfection is possible even in the presence of protective antibodies, particularly in immunocompromised persons [29,32,33,34]. In addition, we identified one PWID with circulating HEV IgM. The minimal degree of hepatitis and negative HEV RNA result in this patient is suggestive of recovered infection. Persistent prolonged HEV IgM positivity has been reported in 24% of patients after acute hepatitis E infection [35]. With all this evidence, we unequivocally conclude that PWIDs in Hong Kong are at higher risk of contracting HEV infection than the general population.

This has major implications, as PWIDs already have a high incidence of liver disease due to alcoholism, chronic hepatitis B and hepatitis C carriage [36]. HEV superinfections in such vulnerable patients could result in acute hepatic decompensation as has been shown in other studies [37,38]. Furthermore, PWIDs are also susceptible to HIV infection, which is a known risk factor for chronic hepatitis E infection [39].

We considered the possibility of other confounding factors causing higher HEV seroprevalence in our PWID cohort. There were more individuals of non-Chinese origin in the PWID group than the donor group, although the difference did not reach statistical significance. These individuals were of South Asian, South East Asian or Hispanic origin, which are HEV hyperendemic areas. However, the HEV seroprevalence among non-Chinese individuals in the PWID group was only 2/9 (22.2%). Even after exclusion of non-Chinese patients, the HEV seroprevalence in the PWID group (40/82) was significantly higher than the donor group (*p* < 0.001), showing that the association was independent of the geographical origin of the patients. Although HCV-HEV co-infections are well described and prior HEV infection may accelerate HCV-related fibrosis, HCV carriage by itself would not independently increase risk of HEV acquisition [32,38]. Although we did not have a detailed dietary history in our study subjects, Hong Kong is an urban society with relatively homogenous food consumption habits. Direct contact with pigs is very uncommon. Homelessness has emerged as a risk factor for hepatitis A, although its role in hepatitis E risk in developed countries is less clear with one major North American study finding no association [16,40]. In Hong Kong, rates of homelessness are low [41], and typically motivated by unaffordable housing rather than addictive injection use. A recent local study found that a monthly household income bracket of 30,000 Hong Kong dollars (3,850 USD) or more was protective against hepatitis E while being born in mainland China was a weak risk factor for hepatitis E seropositivity [30]. In addition, incarceration or medical procedures like hemodialysis may also have an impact on HEV seroprevalence. However, we did not have data on these variables in our cohort. Another limitation of our study was that we did not have details of duration of drug use, type of drugs used, travel history and needle sharing behaviors for individual PWIDs and so were unable to further characterize risk factors for HEV among PWIDs. We cannot exclude that our results only apply to a particular subgroup of at-risk PWIDs engaging in high-risk injection behavior. Investigation of hepatitis E seroprevalence among PWIDs not sharing needles or uninfected by other bloodborne viruses deserves further evaluation.

We relied on the Wantai immunoassay kits for HEV serodiagnosis due to its good sensitivity and specificity [42,43]. However, HCV-infected individuals may have polyclonal B-cell activation rendering their sera prone to non-specific positivity in immunoassays [44]. Our study would have been enriched by a specific confirmatory assay, but this is difficult to fulfil because even HEV immunoblots are known to have poor sensitivity [45]. But the strength of association found in our study is highly suggestive of a genuine link between addictive injection use and hepatitis E.

## 5. Conclusions

In conclusion, PWIDs should be considered a high-risk population for hepatitis E in Hong Kong. Rigorous controlled studies in different geographical areas are required to confirm this finding. As hepatitis E infections may have severe consequences in this vulnerable population, measures to reduce their exposure risk including addiction management, counseling on safe needle practices, needle exchange programs and methadone programs are required. Hepatitis E should be considered in PWIDs with acutely deranged liver function, even in the presence of known hepatitis B or hepatitis C. An effective recombinant hepatitis E vaccine licensed in China should be investigated further in this population [46].

## Figures and Tables

**Figure 1 microorganisms-08-00675-f001:**
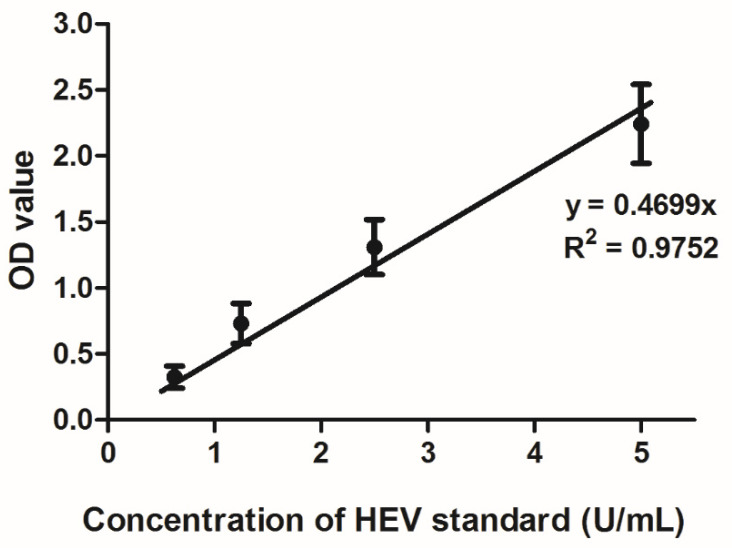
Standard curve of Wantai HEV IgG optical density values against the HEV WHO standard (U/mL). Four concentrations (0.625–5 U/mL) were tested with five replicates per concentration. Bars represent standard error of mean.

**Figure 2 microorganisms-08-00675-f002:**
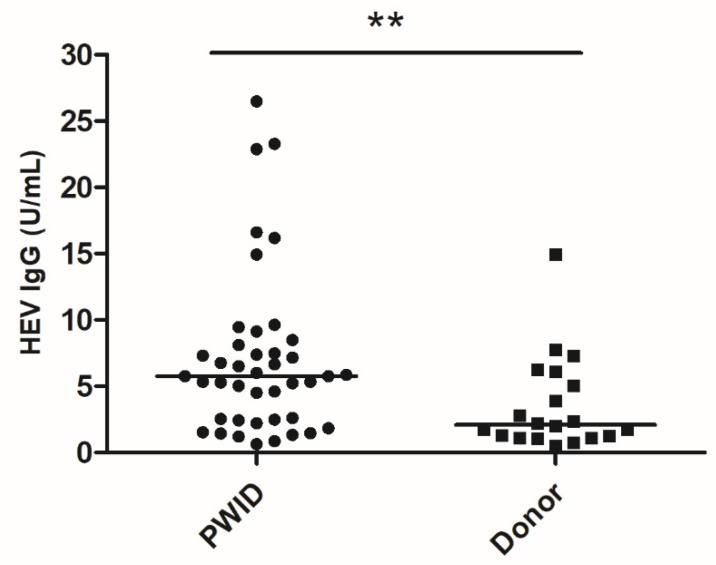
Blood HEV IgG concentrations among PWIDs and organ donors. Bar represents median and (**) indicates statistically significant difference between the median of groups by the Mann–Whitney U test.

**Table 1 microorganisms-08-00675-t001:** Effect of age, gender and addictive injection use on HEV IgG serostatus.

		Bivariable Analysis		Multiple Regression	
Variable	HEV IgG Positive /No. Tested (%)	Odds Ratio ^2^	*p*-Value	Odds Ratio ^2^	*p*-Value
**Age (y)**		1.06 (1.02–1.09) ^1^	<0.005	1.07 (1.03–1.10) ^1^	<0.005
<35	1/11 (9)	0.18 (0.02–1.44)	0.106		
35–44	17/66 (25.8)	0.55 (0.28–1.07)	0.076		
45–54	16/57 (28.1)	0.67 (0.34–1.33)	0.251		
55–64	19/32 (59.3)	3.64 (1.65–8.01)	<0.005		
>65	9/16 (56.3)	2.74 (0.97–7.78)	0.057		
**Sex**					
Male	52/150 (34.7)	1.17 (0.51–2.65)	0.711	NA	NA
Female	10/32 (31.3)	Ref			
**PWID**	42/91 (46.2)	3.04 (1.59–5.79)	<0.005	3.43 (1.72–6.81) ^1^	<0.005
**Donor**	20/91 (22)	Ref			

HEV: hepatitis E virus, PWID: persons who inject drugs, Ref: reference against which the other variable is compared. ^1^ calculated by regression analysis with age as a quantitative explanatory variable. ^2^ figures in parentheses represent a 95% confidence interval.

**Table 2 microorganisms-08-00675-t002:** Previous studies on HEV seroprevalence among PWIDs including control groups.

Reference	Country; HEV IgG Assay	PWID HEV IgG Positive/No. Tested (%)	Control Group	Control Group HEV-IgG Positive/No. Tested (%)	Significant Difference between PWID and Control Groups?
[13]	France; Wantai	22/52 (42.3%)	Blood donors; age- and sex-matched	43/99 (43.4%)	No *p* = 0.890
[14]	Sweden; Diapro	(26%)	HCV patients, but HCV acquired via blood transfusion; not age- and sex-matched	(48%)	Yes, but higher in transfusion-acquired group *p* < 0.020
[15]	Croatia; Euroimmun	3/49 (6.1%)	Healthcare professionals	12/214 (5.6%)	No (*p* > 0.050)
[17]	USA; In-house assay	68/295 (23%)	Blood donors; not age- and sex-matched	64/300 (21.3%)	No (*p* = 0.614)
[18]	Sweden; Abbott	21/34 (62%)	HBV carriers; older control group/ sex-matched	9/36 (25%)	Yes *p* < 0.005
[19]	Italy; Abbott	15/279 (5.4%)	General population; not age- and sex-matched	50/1889 (2.6%)	Yes *p* < 0.050
[20]	Denmark; Abbott	36/137 (26.3%)	Non-drug-using prisoners; not age- and sex-matched	N/A	No (*p* value not shown)
[24]	Brazil; Abbott	12/102 (11.8%)	Blood donor; not age- and sex-matched	4/93 (4.3%)	No (*p* = 0.070)

HEV: hepatitis E virus, PWID: persons who inject drugs, N/A: not available.

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
