# Peer review of "Risk of Hepatitis E among Persons Who Inject Drugs in Hong Kong: A Qualitative and Quantitative Serological Analysis"

_microorganisms, 2020, doi:10.3390/microorganisms8050675_

Round 1
Reviewer 1 Report
In this well written article Sridhar et al. compared HEV IgG seroprevalence among 91 persons who inject drugs (PWIDs) and 91 age and sex matched 91 donors. The purpose of this study is well specified and arises from the fact that in the literature there are conflicting results.
The methodology is correct and the presentation of the data is clear, however there are some missing data:
- the geographical origin of the included patients should be specified because, as expressed by the authors, the epidemiology of infection among PWIDs patients varies from 2.8 to 62% in Europe and North America. Indeed, one might think that PWIDs come from regions with higher prevalence.
- are data available on previous imprisonment of PWIDs patients?
- are there dialysis patients in the PWIDs group? In fact, a Greek study published in Transfusion in 1998 described an increase in HEV seroprevalence in dialysis patients.
- As correctly expressed by the authors, the transmission of HEV is mainly foodborne. Are data on the eating habits of PWDIs patients available? If not, is HAV serology available in these subjects? This could help to characterize a favorable epidemiological set for enterically transmitted infection in the studied populations (Mem Inst Oswaldo Cruz. 2001 Jan;96(1):25-9).
Finally, I agree with the authors that the absence of information regarding the type and duration of drug addiction is a limitation of the study.
Reviewer 2 Report
In the present manuscript, Sridhar et al. have determined the hepatitis E Virus (HEV) IgG seroprevalence in persons who inject drugs (PWIDs) in Hong Kong to investigate the risk of hepatitis E among this population. Overall, this study could be of particular interest for the field but the present manuscript could be improved by addressing further the following points.
Major comments:
- The sample size (91) of the patient cohort remains limited. Is it not possible to increase the number of participants to strengthen the data obtained?
- Could you explain more clearly the importance of IgG quantification? Is a repeated exposure to the virus the only factor that can influence the quantity of IgG detected? From these data, is it really accurate to conclude that “PWIDs are prone to repeated HEV exposure and reinfection as indicated by higher HEV IgG concentrations”?
- Line 63-65: “We evaluated HCV-infected PWIDs because this is an indicator of high-risk practices such as needle sharing [26,27], which in turn would render these individuals at higher risk of other bloodborne infections like hepatitis E”. Is testing HCV-infected PWIDs only appropriate? By doing so, a “higher risk” population is selected that may be not be representative of the general PWID population. It would then be interesting to also include non-HCV-infected PWIDs and compare the different groups.
Minor comments:
- Are data already published on the seroprevalence of HEV antibody in the general population in Hong Kong? It could be interesting to compare these data with the ones found in this study.
- In material and method, it is written “For minipools testing positive for either HEV-A or HEV-C1 RNA, independent samples were extracted and re-tested to identify the positive sample. Sequencing was attempted on positive samples as previously described [3,6]” but later it is stated that “None of the organ donor or PWID blood minipools tested positive for HEV-A or HEV-C1 RNA”. This is quite confusing. Moreover, have you checked first that the sensibility of the assay is not affected by pooling samples?
- Line 183-184: is it not more accurate to conclude that “PWIDs are at higher risk of contracting HEV infection than the general population” in Honk Kong specifically? Could the findings of this study be dependent on the geographical area? Could this also explain (at least partially) that studies on PWIDs have produced contradictory results?
Reviewer 3 Report
The manuscript titled “Risk of hepatitis E among persons who inject drugs in Hong Kong: a qualitative and quantitative 
serological analysis” by Sridhar 
et al. is basically a statistical analysis of the risk of hepatitis E virus infection among the persons injecting drugs. This study is relevant in the aspect that it involves the large sample size than the related studies performed earlier and provides the serological quantification. The manuscript is overall well-structured and findings are interesting. I have couple of minor concerns:
Results:
Line114: Please include data of RT-PCR with known positive control.
Were the population under consideration was also looked for any other liver ailment? Or if they were looked for any comorbidity? Please comment.
Author Response
The manuscript titled “Risk of hepatitis E among persons who inject drugs in Hong Kong: a qualitative and quantitative 
serological analysis” by Sridhar 
et al. is basically a statistical analysis of the risk of hepatitis E virus infection among the persons injecting drugs. This study is relevant in the aspect that it involves the large sample size than the related studies performed earlier and provides the serological quantification. The manuscript is overall well-structured and findings are interesting. I have couple of minor concerns:
Results:
Line114: Please include data of RT-PCR with known positive control.
Reply: Thank you very much for your comment. We have amended the relevant sentence in the results section. Line 117 – 118.
Were the population under consideration was also looked for any other liver ailment? Or if they were looked for any comorbidity? Please comment.
Reply: Thank you. PWIDs were all actively infected with HCV and some of them were also co-infected with hepatitis B. We could not evaluate rates of non-alcoholic fatty liver disease or alcoholism as data was not available for all patients. 3 of the donor group were HbsAg positive, which is quite common in Hong Kong.
Reviewer 4 Report
Sridhar and colleague studied the prevalence of HEV among the persons who inject drugs in Hong Kong. Hepatitis E infection is largely due to feco-oral route and is mostly asymptomatic. HEV infection is problematic in immuno-compromised state.
The overall seropositivity of HEV in Hong Kong is nearly 30%. Furthermore, the prevalence of HEV-antibodies increase with age. These facts are known. The present study has some flows.
- Why authors think that person who inject drugs were NOT already have HEV infection (asymptomatic) before being enrolled in the study.
- What are the evidences that prove that HEV can be transmitted by blood? It is a controversial issue and need to address carefully.
- Was there any subjects that were negative at the start of the study and eventually develop HEV infection?
Minor:
Section 2.3: Total nucleic acid was isolated or RNA was isolated?
Author Response
Sridhar and colleague studied the prevalence of HEV among the persons who inject drugs in Hong Kong. Hepatitis E infection is largely due to feco-oral route and is mostly asymptomatic. HEV infection is problematic in immuno-compromised state.
The overall seropositivity of HEV in Hong Kong is nearly 30%. Furthermore, the prevalence of HEV-antibodies increase with age. These facts are known. The present study has some flows.
- Why authors think that person who inject drugs were NOT already have HEV infection (asymptomatic) before being enrolled in the study.
Reply: Thank you for your question. This study is examining the hypothesis that persons who inject drugs (PWIDs) are more exposed to hepatitis E than the general population. We used HEV IgG, which is a marker of prior exposure to hepatitis E and HEV IgM + RT-PCR, which are markers of active hepatitis E infection to investigate this hypothesis. We do not make the assumption that PWIDs have not already had HEV infection prior to study entry. Instead, whether they have had HEV exposure at study entry is precisely what we are trying to investigate.
2. What are the evidences that prove that HEV can be transmitted by blood? It is a controversial issue and need to address carefully.
Reply: It is now well accepted that hepatitis E is transmissible by blood products in the transfusion setting. Several countries routinely screen blood products for hepatitis E. May we refer the reviewer to citation no. 8 in our manuscript, which shows the extent of bloodborne transmission in the UK. As an extension of this fact, we hypothesized that PWIDs might also be prone to hepatitis E via addictive injection use.
- Was there any subjects that were negative at the start of the study and eventually develop HEV infection?
Reply: Thank you. Please note that this study is not prospective by design. We examined PWID and donor sera only at a single time point as we were not investigating incidence of hepatitis E in PWIDs. A prospective study following up a cohort of PWIDs would be very interesting to answer this question, but it is not the objective of the current study.
Minor:
Section 2.3: Total nucleic acid was isolated or RNA was isolated?
Reply: We isolated total nucleic acid as stated in line 89 – 90.
Reviewer 5 Report
Though this manuscript includes a relatively low number of patients, it is well performed, including a matched control group, and the conclusions are solid, mainly those regarding the IgG concentrations revealing a higher exposure to the virus than the control group.
Minor comments
Introduction
It would be good to add data on the HEV seroprevalence in Hong Kong, for example from blood donors
Methods
Patients: It would be important to know if any of the included HCV-positive patients have previously treated for HCV since RBV and likely some DAAs may be antiviral activity for HEV.
What is the sensitivity of the HEV PCR used for the study?
Statistical analysis
In the logistic regression age was enter as a qualitative factor: this is not mentioned at the methods: are the results the same when this variable is introduced as quantitative or qualitative but as 3 groups?
Results
The IgM-positive patient, was also IgG positive?
Discussion
- Page 6, line 169: adequate sample size: it is not mentioned at the methods if the sample size was calculated prior to the performance of the study.
- The IgM-positive patient has relatively low ALT levels and the HEV RNA was negative, it may worth mention the long-term persistence of HEV IgM after a self-limited hepatitis E infection:
Riveiro-Barciela M, et al. J Viral Hepat. 2020 Feb 27. doi: 10.1111/jvh.13285.
Table 2
It would be interesting to add: the IgG assay used at each of the mentioned studies; complete data on Number of patients included; If RNA was carried out and their result; p value of the last column (significant difference).
Round 2
Reviewer 4 Report
I am still not convinced by the arguments offered by the authors. HEV still a largely a feco-orally transmitted virus. There is no proof offered by the authors that suggests the infection in PWID is not due to the feco-oral infection.
Such findings may bring noise in the field of HEV-epidemiology. I recommend rejection of the manuscript. It will be good from Science as well as Journal's perspective.